# ENJOY YOUR EDITING: CONTROLLABLE GANS FOR IMAGE EDITING VIA LATENT SPACE NAVIGATION

**Peiye Zhuang, Oluwasanmi Koyejo, Alexander G. Schwing**
University of Illinois at Urbana-Champaign
`{peiye, sanmi, aschwing}@illinois.edu`

## ABSTRACT

Controllable semantic image editing enables a user to change entire image attributes with a few clicks, e.g., gradually making a summer scene look like it was taken in winter. Classic approaches for this task use a Generative Adversarial Net (GAN) to learn a latent space and suitable latent-space transformations. However, current approaches often suffer from attribute edits that are entangled, global image identity changes, and diminished photo-realism. To address these concerns, we learn multiple attribute transformations simultaneously, integrate attribute regression into the training of transformation functions, and apply a content loss and an adversarial loss that encourages the maintenance of image identity and photo-realism. We propose quantitative evaluation strategies for measuring controllable editing performance, unlike prior work, which primarily focuses on qualitative evaluation. Our model permits better control for both single- and multiple-attribute editing while preserving image identity and realism during transformation. We provide empirical results for both natural and synthetic images, highlighting that our model achieves state-of-the-art performance for targeted image manipulation.

## 1 INTRODUCTION

Semantic image editing is the task of transforming a source image to a target image while modifying desired semantic attributes, e.g., to make an image taken during summer look like it was captured in winter. The ability to semantically edit images is useful for various real-world tasks, including artistic visualization, design, photo enhancement, and targeted data augmentation. To this end, semantic image editing has two primary goals: (i) providing continuous manipulation of multiple attributes simultaneously and (ii) maintaining the original image's identity as much as possible while ensuring photo-realism.

Existing GAN-based approaches for semantic image editing can be categorized roughly into two groups: (i) *image-space editing* methods directly transform one image to another across domains (Choi et al., 2018; 2020; Isola et al., 2017; Lee et al., 2020; Wu et al., 2019; Zhu et al., 2017a;b), usually using variants of generative adversarial nets (GANs) (Goodfellow et al., 2014). These approaches often have high computational cost, and they primarily focus on binary attribute (on/off) changes, rather than providing continuous attribute editing abilities. (ii) *latent-space editing* methods focus on discovering latent variable manipulations that permit continuous semantic image edits. The chosen latent space is most often the latent space of GANs. Both unsupervised and (self-)supervised latent space editing methods have been proposed. Unsupervised latent-space editing methods (Härkönen et al., 2020; Voynov & Babenko, 2020) are often less effective at providing semantically meaningful directions and all too often change image identity during an edit. Current (self-)supervised methods (Jahanian et al., 2019; Plumerault et al., 2020) are limited to geometric edits such as rotation and scale. To our knowledge, only one supervised approach has been proposed (Shen et al., 2019) – developed to discover semantic latent-space directions for binary attributes. As we show, this method suffers from entangled attributes and often does not preserve image identity during manipulation.

**Contributions.** We propose a latent-space editing framework for semantic image manipulation that fulfills the aforementioned primary goals. Specifically, we use a GAN and employ a joint sampling strategy trained to edit multiple attributes simultaneously. To disentangle attribute transformations

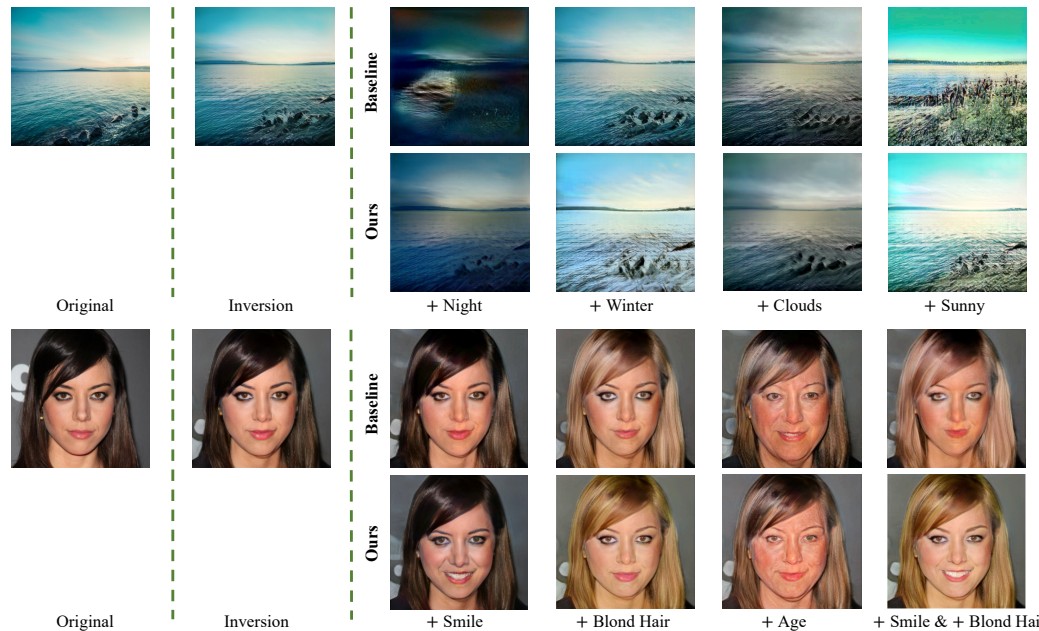

Figure 1: **Real image manipulation** on scene (top two rows, photo from Flickr) and face (bottom two rows, unseen image from CelebA-HQ) using pretrained StyleGAN2 (Karras et al., 2019b): We reconstruct the real images (col.1) by finding a latent vector with the best inversion result (col.2) on StyleGAN2 (Abdal et al., 2019). After that, we transform the latent vectors for single- and multiple-attribute manipulations (col.3-6). Note that unlike ours, the baseline method (Shen et al., 2019) either changes image identity or confounds semantic properties, or both.

in the latent space of GANs, we integrate a regressor to predict the attributes that an image exhibits. The regressor also permits precise control of the manipulation degree and is easily extended to multiple attributes simultaneously. In addition, we incorporate a perceptual loss (Li et al., 2019) and an adversarial loss that helps preserve image identity and photo-realism during manipulation.

We compare our method to several popular frameworks, from existing image-to-image translation methods (Choi et al., 2020; Wu et al., 2019; Zhu et al., 2017a) to latent space transformation-based approaches (Shen et al., 2019; Voynov & Babenko, 2020). We mention that prior work primarily uses qualitative evaluation like the one in Fig. 1. In contrast, we propose a quantitative evaluation to measure *controllability*. Both qualitative and quantitative results provide evidence that our approach outperforms prior work in terms of quality of the semantic image manipulation while maintaining image identity.

## 2 RELATED WORK

**Generative Adversarial Networks (GANs)** (Goodfellow et al., 2014) have significantly improved realistic image generation in recent years (Brock et al., 2018; Jolicoeur-Martineau, 2019; Karras et al., 2017; 2019a;b; Park et al., 2019; Zhang et al., 2018). For this, a GAN formulates a 2-player non-cooperative game between two deep nets: (i) a generator that produces an image given a random noise vector in the latent space, sampled from a known prior distribution, usually a normal or a uniform distribution; (ii) a discriminator whose input is both synthetic and real data, which is to be differentiated.

**Semantic image editing** seeks to automate image manipulation of semantics. Encouraged by the success of deep nets, recent works have applied deep learning methods for semantic image editing tasks such as style transfer (Li et al., 2017; Luan et al., 2017), image-to-image translation (Choi et al., 2018; 2020; Isola et al., 2017; Lee et al., 2020; Wang et al., 2018; Wu et al., 2019; Zhu et al., 2017a) and discovering semantically meaningful directions in a GAN latent space (Härkönen et al., 2020; Jahanian et al., 2019; Plumerault et al., 2020; Shen et al., 2019; Voynov & Babenko, 2020). Note that our task is an extended version of semantic image editing that requires more comprehensive control to satisfy user-desired operations. Therefore, most aforementioned approaches do not meet

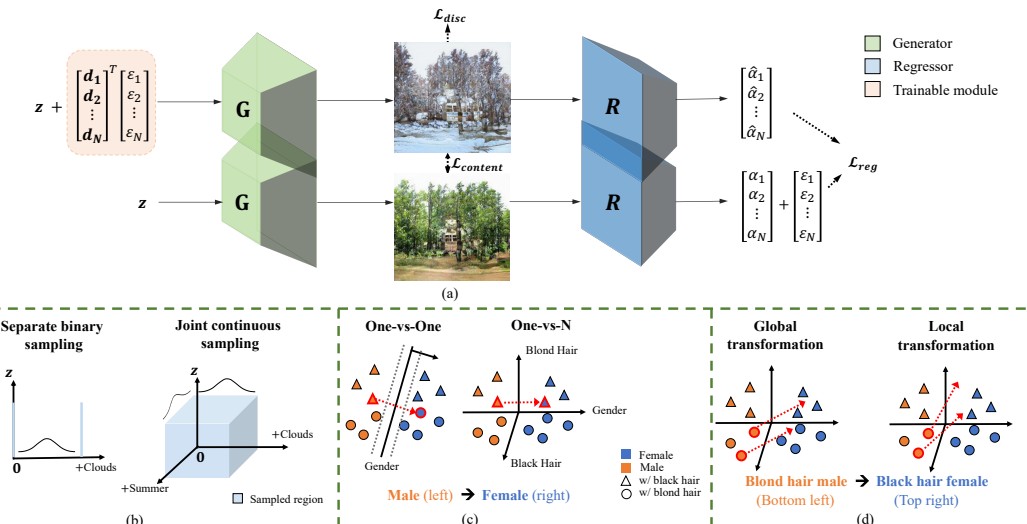

Figure 2: **Our overall framework** (top row) and 3 training strategies different from prior work (bottom row). In (a), $G$ takes $z$ and an edited latent vector separately to synthesize images. $T = \{d_1, ..., d_N\}$ are the trainable latent-space directions and $\varepsilon$ represents transformation extent. Original and edited image attributes, $\alpha$ and $\hat{\alpha}$, are predicted by the pre-trained regressor $R$. The discriminator loss $\mathcal{L}_{\text{disc}}$ (the discriminator $D$ not shown due to limited space), the regression loss $\mathcal{L}_{\text{reg}}$, and the perceptual loss $\mathcal{L}_{\text{content}}$ are used to update $T$. We explain (b-d) in Sec. 3.2.

the requirements. Nonetheless, we categorize the approaches most relevant to our task into two groups: (i) image editing via manipulation in image space, and (ii) image editing via latent space navigation.

**Image-space editing using GANs** directly manipulates an image for targeted editing. Early work (Isola et al., 2017) used GANs to implement semantic translations between two image domains with paired data, e.g., day to night. Follow-up work focused on multi-modal (Bhattarai & Kim, 2020; Choi et al., 2018; 2020; Lee et al., 2020; Wang et al., 2018; Wu et al., 2019) and unpaired image domain translation (Zhu et al., 2017a). In this case, they primarily consider binary (on/off) attribute changes, regardless of whether the process is dynamic, for example, day to night.

**Latent-space editing via GANs** has received an increasing amount of recent interest. Most prior work focused on identifying semantically meaningful directions in the latent space of GANs, so that shifting latent vectors towards these directions achieves the desired image manipulation. Recent papers found semantics in the latent space of GANs, such as color transformations and camera movements (Jahanian et al., 2019), or face attribute changes (Shen et al., 2019), such as smile. Other work considered unsupervised methods (Härkönen et al., 2020; Voynov & Babenko, 2020) to discover interpretable latent space directions. However, additional challenges inherent to unsupervised manipulation of the latent space arise, and have not been addressed in prior work, e.g., direction quality with regard to degree control and image identity preservation.

## 3 METHOD

### 3.1 PROBLEM STATEMENT

We consider controllable semantic image editing via latent space navigation in GANs. We begin with a *fixed* GAN model that consists of a generator $G$ and a discriminator $D$. The input of $G$ is a latent vector $z \in \mathbb{R}^m$ from a latent space $\mathcal{Z}$. Here, $m$ is the dimensionality of the latent space. Given $N$ attributes, we aim to discover semantically meaningful latent-space GAN directions, $T = \{d_1, \ldots, d_N\}$, to manipulate the attributes of synthetic images $G(z)$ with an assigned step size $\varepsilon = \{\varepsilon_1, \ldots, \varepsilon_N\}$, where $d_i \in \mathbb{R}^m$ for all $i \in \{1, \ldots, N\}$. In the end, each attribute of an edited image can be changed with the corresponding degree $\varepsilon$ from $\alpha = \{\alpha_1, \ldots, \alpha_N\}$, the original attributes of $G(z)$ predicted by a regressor $R$.

## 3.2 Proposed approach

We provide an overview of our method in Fig. 2 (a), and further illustrate how our approach differs from prior work in Fig. 2 (b-d).

**Overview.** As shown in Fig. 2 (a), we employ a GAN model consisting of a generator $G$ and a discriminator $D$ (not shown due to limited space), as well as a regressor $R$, all of which are pre-trained. Our goal is to find latent directions $\boldsymbol{T}$ that provide attribute specific image edits. At each training step, $G$ takes $\boldsymbol{z}$ and an edited latent vector referred to as $\boldsymbol{z}'$. We follow prior work which suggests that direction vectors in latent space of GANs permit image editing (Jahanian et al., 2019). Formally, given a transformation degree $\boldsymbol{\varepsilon}$, we obtain the latent-space edit as $\boldsymbol{z}' = \boldsymbol{z} + \boldsymbol{T}\boldsymbol{\varepsilon} = \boldsymbol{z} + \sum_{i=1}^{N} \varepsilon_i \boldsymbol{d}_i$. $G$ provides the recovered and edited images $G(\boldsymbol{z})$ and $G(\boldsymbol{z}')$, which are separately processed by the regressor $R$ to predict attribute values. The original attributes of $G(\boldsymbol{z})$ are $\boldsymbol{\alpha} = R(G(\boldsymbol{z}))$. The (pseudo-) ground truth and predicted attribute of $G(\boldsymbol{z}')$ are $\boldsymbol{\alpha}'$ and $\hat{\boldsymbol{\alpha}}'$, respectively, where $\boldsymbol{\alpha}' = \boldsymbol{\alpha} + \boldsymbol{\varepsilon}$ and $\hat{\boldsymbol{\alpha}}' = R(G(\boldsymbol{z}'))$.

Intuitively, the goal of $\boldsymbol{T}$ is to transform $\boldsymbol{z}$ by adding semantically meaningful information such that the corresponding output image $G(\boldsymbol{z}')$ exhibits attribute changes $\boldsymbol{\varepsilon}$ from $\boldsymbol{\alpha}$. In practice, we normalize the range of attribute values to $[\boldsymbol{0}, \boldsymbol{1}]$, i.e., both $\boldsymbol{\alpha}$ and $\boldsymbol{\alpha}' \in [\boldsymbol{0}, \boldsymbol{1}]$. During training, we maintain the unit range by controlling the given $\boldsymbol{\varepsilon}$. Formally, $\boldsymbol{\varepsilon}$ is drawn from a distribution $\mathcal{D}_\varepsilon$ uniform in $[-1, 1]^N$ while considering the constraint that $\boldsymbol{0} \leq \boldsymbol{\alpha} + \boldsymbol{\varepsilon} \leq \boldsymbol{1}$.

**Objective function.** To find $\boldsymbol{T}$ we minimize the weighted objective:

$$\min_{\boldsymbol{T}} \mathcal{L} \triangleq \lambda_1 \mathcal{L}_{\text{reg}} + \lambda_2 \mathcal{L}_{\text{disc}} + \lambda_3 \mathcal{L}_{\text{content}}. \tag{1}$$

Note, the objective is only used for optimizing $\boldsymbol{T}$, while the other modules remain fixed. Hyperparameters $\lambda_1, \lambda_2$, and $\lambda_3$ adjust the loss terms.

The regression loss $\mathcal{L}_{\text{reg}}$ assesses whether $\boldsymbol{T}$ performs the transformations indicated by $\boldsymbol{\varepsilon}$. We express the regression loss via a binary cross entropy:

$$\mathcal{L}_{\text{reg}} = \quad \mathbb{E}_{\boldsymbol{z} \sim \mathcal{Z}, \boldsymbol{\varepsilon} \sim \mathcal{D}_\varepsilon} [-\hat{\boldsymbol{\alpha}}' \log \boldsymbol{\alpha}' - (\boldsymbol{1} - \hat{\boldsymbol{\alpha}}') \log (\boldsymbol{1} - \boldsymbol{\alpha}')]. \tag{2}$$

Note that $\boldsymbol{\alpha}'$ is from the distribution generated by $\boldsymbol{z}$ and $\boldsymbol{\varepsilon}$. Please refer to the appendix C.1 for more details on this distribution.

The second loss term $\mathcal{L}_{\text{disc}}$ is computed using the discriminator $D$ and measures the quality of the generated images, i.e.,

$$\mathcal{L}_{\text{disc}} = \quad \mathbb{E}_{\boldsymbol{z}' \sim \mathcal{Z}'|\boldsymbol{z}} \left[ \log(1 - D(G(\boldsymbol{z}'))) \right]. \tag{3}$$

Here we refer to the domain of $\boldsymbol{z}'$ via $\mathcal{Z}'$, which is conditioned on $\boldsymbol{z}$. We write this using $\mathcal{Z}'|\boldsymbol{z}$.

Lastly, we use a content loss $\mathcal{L}_{\text{content}}$, often also referred to as perceptual loss. It is designed to estimate the distance between two images and it is employed to maintain the image identity during the transformation. Specifically, we use the content loss term

$$\mathcal{L}_{\text{content}} = \quad \mathbb{E}_{\boldsymbol{z} \sim \mathcal{Z}, \boldsymbol{z}' \sim \mathcal{Z}'|\boldsymbol{z}} \sum_{i \in \mathcal{D}_{\text{content}}} \|F_i(G(\boldsymbol{z}')) - F_i(G(\boldsymbol{z}))\|_2^2, \tag{4}$$

where $F_i(\cdot)$ denotes a feature function which extracts intermediate features from images. $\mathcal{D}_{\text{content}}$ indicates the layers of a pre-trained model which are used as features. We approximate the aforementioned expectations by empirical sampling. We defer algorithm details to Appendix A.

**Joint-distribution sampling and training.** The regressor operates in a multi-label setting, i.e., each data possesses multiple attributes. To ensure the disentanglement of attribute edits we sample synthetic images from the entire data distribution and find all transformations at once (illustrated in Fig. 2 (b,c) right). In contrast, Shen et al. (2019) prepare training samples on the two opposing data subsets with regards to an attribute, e.g., no clouds *vs.* many clouds, and find the directions with $N$ one-vs-one classifiers (sketched in Fig. 2 (b,c) left).

**Transformation module $\boldsymbol{T}$.** We study two types of transformations $\boldsymbol{T}$: (i) *global* and (ii) *local*. A *global* transformation $\boldsymbol{T}$ refers to a semantic latent-space transformation identical for all $\boldsymbol{z}$ during inference. This is illustrated via parallel red dashed arrows in Fig. 2 (d) left. These global directions

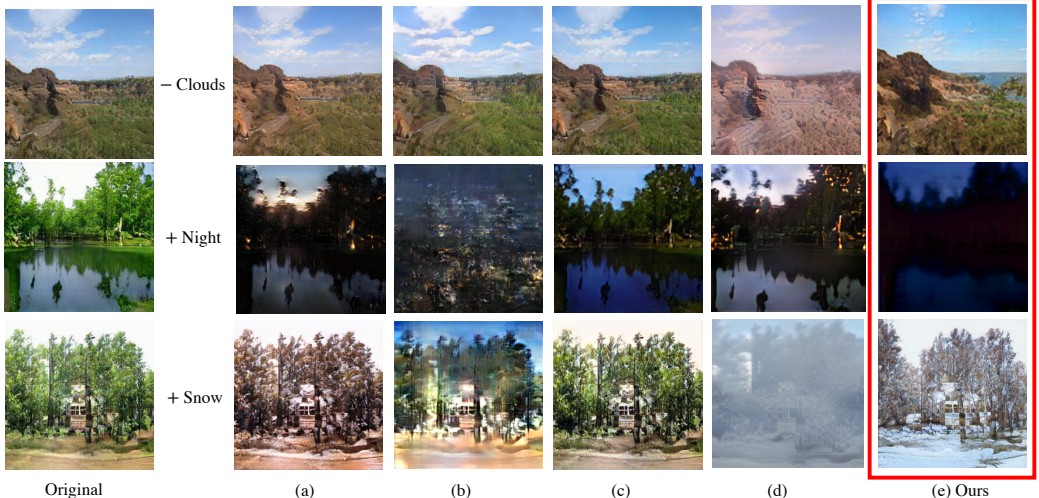

Figure 3: **Comparison of image-space editing approaches** with respect to "removing clouds" (top), "enhancing night" (middle) and "adding snow" (bottom). The original images (col.1) are followed by a given editing task. From (a-e): (a) CycleGAN (Zhu et al., 2017a), (b) StarGAN v2 (Choi et al., 2020), (c) RelGAN (Wu et al., 2019), (d) DRIT++ (Lee et al., 2020), and (e) Ours.

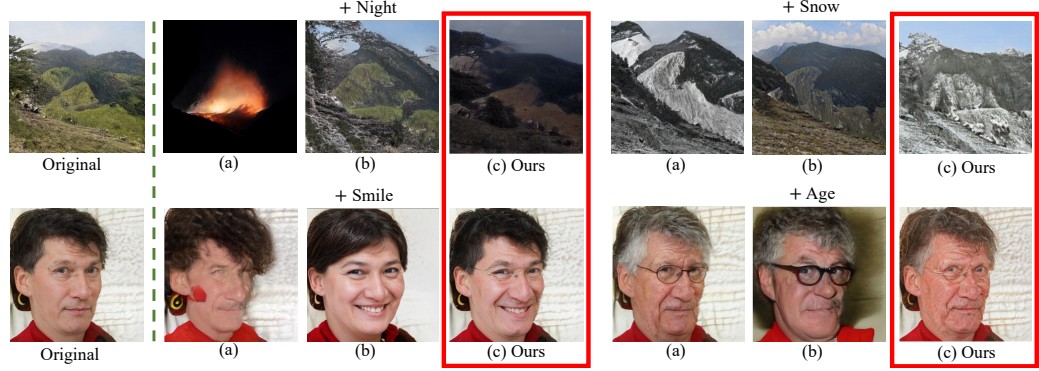

Figure 4: **Comparison of latent-space editing approaches on StyleGAN2.** (a) Shen et al. (2019); (b) Voynov & Babenko (2020); (c) Ours. Original synthetic image (col.1); edited results of a semantic manipulation noted on top (col.2-4; col.5-7). Shen & Zhou (2021) mention that "age," "gender," and "glasses" directions are hard to disentangle potentially due to data bias, while our results suggest a better direction disentanglement.

are commonly used (Härkönen et al., 2020; Jahanian et al., 2019; Shen et al., 2019; Viazovetskyi et al., 2020). However, a globally identical direction might not serve all data. In contrast, a *local* transformation $T$ is a function of $z$ which provides various directions for different $z$ (in Fig. 2 (d) right, shown via non-parallel red arrows). Formally, $d_i = f_\theta^i(z)$, where $f_\theta^i$ is implemented via a deep net. In Sec. 4 we show that the local transformation $T$ succeeds on failure cases of the global transformation.

## 4 EXPERIMENTS

We introduce datasets, implementation details, evaluation metrics, and show results next.

**Datasets.** We evaluate the proposed approach on two types of datasets: (i) face datasets – FFHQ (Karras et al., 2019a), CelebA (Liu et al., 2018) and CelebA-HQ (Karras et al., 2017), commonly used in prior work (Härkönen et al., 2020; Karras et al., 2017; 2019a;b; Shen et al., 2019;

Viazovetskyi et al., 2020). (ii) natural scene datasets – Transient Attribute Database (Laffont et al., 2014) and MIT Places2 data (Zhou et al., 2017), which contain attributes suitable for continuous semantic image editing. We briefly introduce the scene datasets:

- *Transient Attribute Database* (Laffont et al., 2014): It contains 8,571 scene images with 40 attributes in 5 categories including lighting (e.g., "bright"), weather (e.g., "cloudy") , seasons (e.g., "winter"), subjective impressions (e.g., "beautiful"), and additional attributes (e.g., "dirty"). Each attribute is annotated with a real-valued score between $0$ and $1$, where $0$ indicates absence of the attribute.
- *MIT Places2 data* (Zhou et al., 2017): Using the provided category annotations (i.e., indoor/outdoor and natural/artificial), we select the natural outdoor scenes, obtaining a total of 144,543 images.

**Implementation details.** We choose $\lambda_1 = 10, \lambda_2 = 0.05, \lambda_3 = 0.05$ in Eq. (1) and compute the perceptual loss using the conv1_2, conv2_2, conv3_2, conv4_2 activations in a VGG-19 network (Simonyan & Zisserman, 2014) pre-trained on the ImageNet dataset (Russakovsky et al., 2015). We train $T$ for 50k iterations with a batch size of 4. An Adam optimizer is used with a learning rate of 1e-4. For the latent space $\mathcal{Z}$ in the proposed method we use the $\mathcal{W}$ space of StyleGAN2 and the $\mathcal{Z}$ space of PGGAN. For the attribute regressor $R$, we adopt a ResNet-50 (He et al., 2016) for attribute prediction on the Transient Attribute (Laffont et al., 2014) and the CelebA (Liu et al., 2018) data. The last fully connected layer in the ResNet-50 is replaced by a linear layer with an output dimension of 40. We train the regressors for 500 epochs and use the weights with the best validation mean squared error (MSE) on CelebA and the best test MSE on the Transient Scene Database. The GAN follows the StyleGAN2 architecture and is pretrained with 200k iterations on a union of the two natural scene datasets using a resolution of $256 \times 256$. The training batch size is 32 and an Adam optimizer is used with a learning rate of 2e-3. For the face datasets, we use FFHQ pre-trained weights[1] of StyleGAN2, and CelebA-HQ weights of PGGAN[2].

**Baselines.** We compare our approach to several popular image-to-image translation approaches (Choi et al., 2020; Wu et al., 2019; Zhu et al., 2017a) and latent space direction discovery methods (Shen et al., 2019; Voynov & Babenko, 2020). Since Choi et al. (2020); Shen et al. (2019); Zhu et al. (2017a) cannot deal with continuous attributes, we split the data into 2 domains using a threshold value of $0.5$ for each of the 40 attributes. Afterward, the models are trained on 2 contrast domains, e.g., with or without a certain attribute. We use the official code of all baseline methods and rigorously follow their training steps. Note that work by Voynov & Babenko (2020) is unsupervised, i.e., the latent-space directions are human interpreted. To avoid bias during the selection of directions, we use the attribute regressor to automatically identify the most significant directions that can edit predetermined attributes. The details of preparing the baselines are given in Appendix C.2.

**Evaluation metrics.** There is no good numerical metric to evaluate image editing (Shen et al., 2019; Voynov & Babenko, 2020). In an attempt to address this concern, we automate quantitative evaluation based on a property that editing of attributes should maintain image identity. To achieve this, we employ a popular image identity recognition model[3] pre-trained on the VGGface2 data (Cao et al., 2018). Cosine similarity is used to represent the similarity between paired original and edited images (Cao et al., 2018). In addition, we evaluate changes of the other independent attributes using the pre-trained regressor.

## 4.1 RESULTS ON NATURAL SCENE DATASETS

**Comparison to image-to-image translation:** As shown in Fig. 3 (a-d), we observe image-to-image translation to perform poorly when editing image details (e.g., removing clouds). Moreover, sometimes artifacts (orange spots in (a) and (b)) are introduced. In contrast, our model shows overall compelling transfer performance on all these attributes.

**Comparison to latent-space translation:** Inspecting Fig. 1, we observe our approach to improve upon work by Shen et al. (2019) in attribute edit quality and number of artifacts. Similarly, in

---

[1]https://github.com/rosinality/stylegan2-pytorch

[2]https://pytorch.org/hub

[3]https://github.com/ox-vgg/vgg_face2

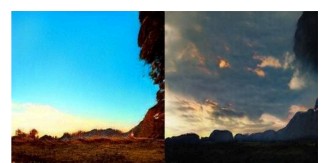
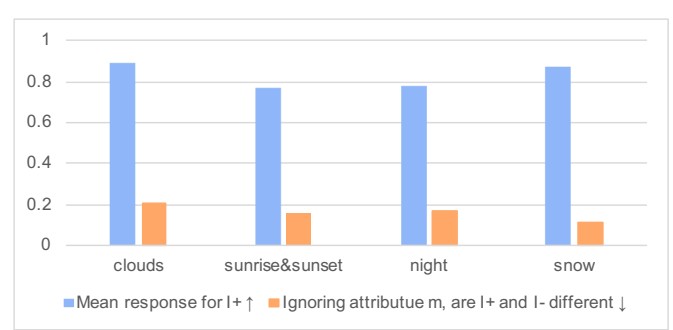

*Question1*: Compared to the picture on the left, how cloudy is the picture on the right?

*Question2*: Ignoring the attribute clouds, determine if the two pictures record the same scene.

Figure 5: **User study for image editing and image identity preservation**. Blue bars show the predicted presence of a given attribute for images ($I+$, higher is better). The orange bars measure the difference between paired images (lower is better).

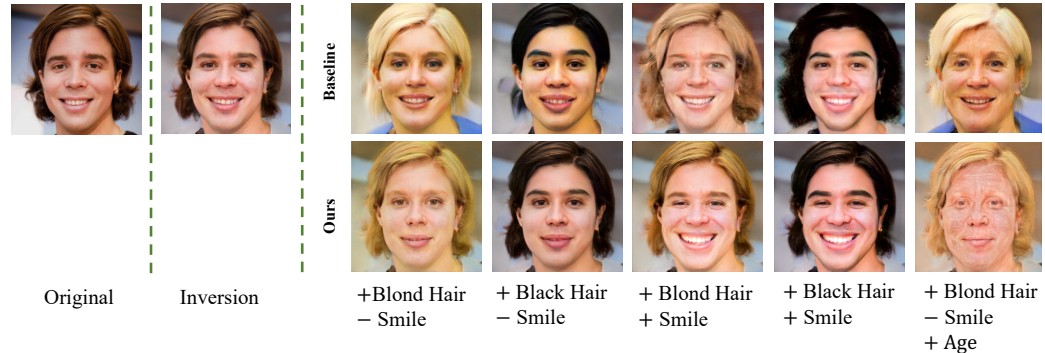

Figure 6: **Real image synthesis with multiple attributes.** Shen et al. (2019) (top row); ours (bottom row): original real image (col.1); inverted result (col.2) from Abdal et al. (2019); edited results (col.3-7) with target attribute changes shown at the bottom.

Fig. 4, we notice that our method performs well in attribute editing, and has a better ability to preserve image identity when adding "night" and "snow." We show additional analysis and results in Appendix D and E.1.

**Quantitative evaluation:** We follow a user study proposed by Kowalski et al. (2020) to evaluate controllability of our scene edits. Concretely, given an image $I = G(z)$, we generate paired images $(I+, I-)$ that have opposing values for attribute $n$. 40 users took the test to answer: (a) the presence and strength of the attribute $n$ in $I+$, and (b) if $I+$ and $I-$ are identical when ignoring the attribute. Both types of questions have options on a 5-level scale from $0$ ("not at all") to $1$ ("totally"). Ideally, the response to question (a) should be $1$ while we expect replies for (b) to be $0$. We test performance of image editing with "clouds", "sunrise&sunset", "night" and "snow" attributes. A question example and statistical results over 50 image pairs are shown in Fig. 5. Statistical evidence supports our claim that the proposed approach performs well with regards to image edits while maintaining image identity.

## 4.2 RESULTS ON FACE DATASETS

**Manipulation results on StyleGAN2:** The comparison on real images shown in Fig. 1 suggests that our method works well for attribute edits. Further, the synthetic image edit results in Fig. 4 indicate that our edits are disentangled, while the baselines unexpectedly add "glasses" when aging the face. To increase the task difficulty, we edit real images with multiple attribute changes simultaneously. Results are summarized in Fig. 6, which highlights the controllability of our edits.

**Quantitative evaluation:** We measure the changing degrees of the other independent attributes and the image identity when editing attributes with various degrees $\hat{\varepsilon}$. Here $\hat{\varepsilon}$ is a predicted changing degree on a target attribute by $R$. For a thorough comparison, we evaluate the performance on 3 seg-

| $\lvert\hat\varepsilon\rvert$ | Smile | | | Hair color | | | Smile + Hair color | | |
|---|---|---|---|---|---|---|---|---|---|
| | (0, .3] | (.3, .6] | (.6, .9] | (0, .3] | (.3, .6] | (.6, .9] | (0, .3] | (.3, .6] | (.6, .9] |
| Shen et al. | .202 | .204 | .224 | .256 | .272 | .277 | .299 | .318 | .329 |
| | ± 5e-2 | ± 6e-2 | ± 4e-2 | ± 1e-2 | ± 2e-2 | ± 2e-2 | ± 5e-3 | ± 1e-2 | ± 3e-2 |
| Voynov et al. | .115 | .211 | .277 | .162 | .166 | .177 | .155 | .220 | .284 |
| | ± 6e-3 | ± 4e-2 | ± 7e-3 | ± 1e-2 | ± 3e-2 | ± 9e-3 | ± 4e-3 | ± 4e-2 | ± 2e-2 |
| **Ours** | **.085** | **.084** | **.098** | **.075** | **.083** | **.084** | **.088** | **.111** | **.134** |
| | ± 4e-2 | ± 3e-3 | ± 4e-3 | ± 7e-4 | ± 5e-3 | ± 5e-3 | ± 7e-3 | ± 4e-3 | ± 4e-3 |

Table 1: **Quantitative evaluation of numerical changes on the other semantically independent attributes** (lower is better). (*first row*) edited attributes; (*second row*) the absolute changing range of the edited attributes; (*bottom three rows*) averages (up) and standard deviations (down) in each row.

| $\lvert\hat\varepsilon\rvert$ | Smile | | | Hair color | | | Smile + Hair color | | |
|---|---|---|---|---|---|---|---|---|---|
| | (0, .3] | (.3, .6] | (.6, .9] | (0, .3] | (.3, .6] | (.6, .9] | (0, .3] | (.3, .6] | (.6, .9] |
| Shen et al. | .918 | .916 | .907 | .887 | .877 | .874 | .877 | .819 | .801 |
| | ± 4e-3 | ± 1e-3 | ± 5e-3 | ± 3e-2 | ± 4e-2 | ± 4e-2 | ± 6e-3 | ± 3e-2 | ± 4e-2 |
| Voynov et al. | .979 | .896 | .869 | .955 | .909 | .904 | .940 | .829 | .811 |
| | ± 2e-3 | ± 4e-3 | ± 9e-3 | ± 6e-2 | ± 4e-2 | ± 3e-2 | ± 9e-3 | ± 4e-2 | ± 3e-2 |
| **Ours** | **.995** | **.994** | **.992** | **.993** | **.993** | **.993** | **.992** | **.986** | **.984** |
| | ± 7e-4 | ± 1e-3 | ± 9e-4 | ± 5e-4 | ± 1e-4 | ± 2e-4 | ± 9e-4 | ± 2e-3 | ± 4e-4 |

Table 2: **Quantitative evaluation on image identity preservation** (higher is better). Notation is identical to Tab. 1.

ments according to the absolute value of $\hat\varepsilon$, i.e., $\lvert\hat\varepsilon\rvert$ in the range of $(0, 0.3]$, $(0.3, 0.6]$ and $(0.6, 0.9]$. Evaluation on "smile," "hair color," and "smile+hair color" attributes are shown in Tab. 1 and Tab. 2. The "hair color" attribute includes "blond" and "black" colors where we average the results on both cases. For multiple attribute editing, "smile + hair color," we evaluate the case that both the two targeted attributes change within the $\lvert\hat\varepsilon\rvert$ range. We use around 1k original images, generate 10k edited images with regard to each target attribute, and repeat the experiment 3 times. The averaged results and the standard deviations presented in Tab. 1 and Tab. 2 suggest that our model outperforms the baselines with regard to disentanglement and image identity preservation.

**Manipulation results on PGGAN:** For PGGAN, we use local transformations parameterized by three-layer perceptrons with leaky ReLU activations. We show the detailed MLP architecture in Appendix B.1. Editing results in Fig. 7 suggest that the proposed approach is able to continuously edit faces with various semantic attributes. We also study the differences between local and global directions in Fig. 11. Particularly, a global $\boldsymbol{T}$ (in Fig. 11 (a)) is a matrix where each column $i$ is given by $\boldsymbol{d}_i$. For a local $\boldsymbol{T}$ (in Fig. 11 (b)) $\boldsymbol{d}_i$ is given by a fully-connected layer with identical input and output dimensions. Fig. 11 (c) and (d) use the MLP components with the aforementioned structure. We use $\mathcal{L}_{\text{reg}}$ to train $\boldsymbol{T}$ in (c), while the total loss $\mathcal{L}$ is used for (d). The results in Fig. 11 indicate that a local $\boldsymbol{T}$, with either linear or MLP structure works well for face attribute edits. Moreover, our loss $\mathcal{L}$ stabilizes the training process of $\boldsymbol{T}$.

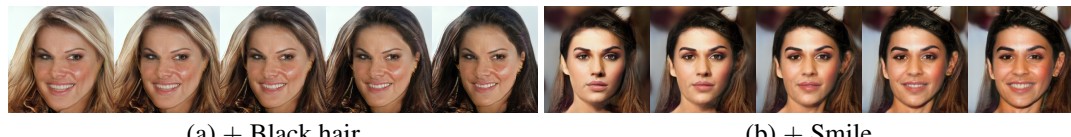

(a) + Black hair          (b) + Smile

Figure 7: **Continuous editing results on PGGAN** (Karras et al., 2017) with the "black hair" (left) and the "smile" (right) attribute.

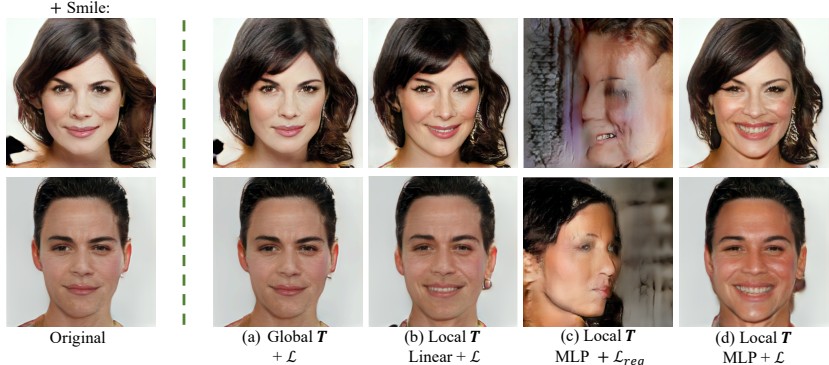

Figure 8: **Comparison of global and local transformations on PGGAN.** Original synthetic images (col.1); the edited results on enhancing "smile" (col.2-5): (a) a global $T$, (b) a local $T$, with linear-layer direction functions, (c) a local $T$ with MLP direction functions trained via $\mathcal{L}_{\text{reg}}$ loss only, and (d) a local $T$ with MLP direction functions training via the entire loss $\mathcal{L}$.

## 5 CONCLUSION

We propose an effective approach to semantically edit images by transferring latent vectors towards meaningful latent space directions. The proposed method enables continuous image manipulations with respect to various attributes. Extensive experiments highlight that the model achieves state-of-the-art performance for targeted image manipulation.

**Acknowledgements:** This work is supported in part by NSF under Grant No. 1718221, 2008387, 1934986 and MRI #1725729, NIFA award 2020-67021-32799, UIUC, Samsung, Amazon, 3M, and Cisco Systems Inc. (Gift Award CG 1377144). We thank Cisco for access to the Arcetri cluster. We thank Amazon for EC2 credits.

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

## APPENDIX

### A.    ALGORITHM

The overall procedure of the proposed method in Algorithm 1, taking local transformation $\boldsymbol{T}$ as an example.

---

**Algorithm 1** Training Procedure

---

**Input:** A pre-trained GAN with $G$, $D$, and input noise distribution $\boldsymbol{z} \sim \mathcal{Z}$; a pre-trained regressor $R$ for predictions on N attributes; an initialized $\boldsymbol{T}$; max iteration number $M$.

1: **for** iteration $m = 1, \ldots, M$ **do**
2:      Sample random noise $\boldsymbol{z} \sim \mathcal{Z}$, and $\boldsymbol{\varepsilon}$
3:      Compute internal attributes for a synthetic image, i.e., $\boldsymbol{\alpha} = R(G(\boldsymbol{z}))$
4:      Compute the actual shift value $\boldsymbol{\delta} = CLIP(\boldsymbol{\alpha} + \boldsymbol{\varepsilon}, (0, 1)) - \boldsymbol{\alpha}$
5:      Compute the transformed latent vector $\boldsymbol{z}' = \boldsymbol{z} + \boldsymbol{T}(\boldsymbol{z})\boldsymbol{\delta}$
6:      Compute $I' = G(\boldsymbol{z}')$, $\boldsymbol{\alpha}' = \boldsymbol{\alpha} + \boldsymbol{\delta}$
7:      Compute attribute predictions $\hat{\boldsymbol{\alpha}}' = R(I')$
8:      Compute the loss $\mathcal{L}$
9:      Update $\boldsymbol{T}$
10: **end for**
**Return:** $\boldsymbol{T}$

---

### B.    METHOD DETAILS

#### B.1.    METHOD ON PGGAN

We apply a multi-layer perceptron (MLP) to parameterize a latent space path on PGGAN. Fig. 9 shows the MLP architecture that we use.

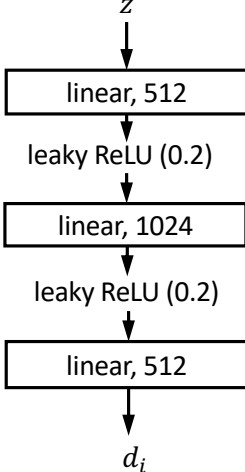

Figure 9: **Example MLP architecture on PGGAN.**

In practice, we find that normalizing the direction vector $\boldsymbol{d}_i$ helps preserve image identity during manipulation on PGGAN. Formally, the transformed latent vector can be written as

$$\boldsymbol{z}' = \boldsymbol{z} + \lambda\boldsymbol{\varepsilon}\frac{\boldsymbol{d}_i}{\|\boldsymbol{d}_i\|}, \tag{5}$$

where $\lambda$ is a weight adjusting the direction scale. We set $\lambda = 3$.

## C.    EXPERIMENTAL DETAILS

### C.1 THE EXPECTATION IN $\mathcal{L}_{\text{REG}}$

We define the regression loss $\mathcal{L}_{\text{reg}}$ as

$$\mathcal{L}_{\text{reg}} = \quad \mathbb{E}_{\boldsymbol{z} \sim \mathcal{Z}, \boldsymbol{\varepsilon} \sim \mathcal{D}_\varepsilon}[-\hat{\boldsymbol{\alpha}}' \log \boldsymbol{\alpha}' - (\boldsymbol{1} - \hat{\boldsymbol{\alpha}}') \log (\boldsymbol{1} - \boldsymbol{\alpha}')], \tag{6}$$

where $\boldsymbol{\alpha}'$ is from the distribution generated by $\boldsymbol{z}$ and $\boldsymbol{\varepsilon}$. We now discuss how we derive the distribution of $\boldsymbol{\alpha}'$. To begin, $\boldsymbol{z}$ is sampled from $\mathcal{Z}$ and $\boldsymbol{\varepsilon}$ is sampled from $\mathcal{D}_\varepsilon$. The other variables are generated by $\boldsymbol{z}$ and $\boldsymbol{\varepsilon}$:

$$\boldsymbol{z}' = \boldsymbol{z} + \boldsymbol{T}\boldsymbol{\varepsilon}$$
$$\boldsymbol{\alpha}' = R(G(\boldsymbol{z})) + \boldsymbol{\varepsilon}$$
$$\hat{\boldsymbol{\alpha}}' = R(G(\boldsymbol{z}')).$$

As a result, the distribution of $\boldsymbol{\alpha}'$ is dependent on $\mathcal{Z}$ and $\mathcal{D}_\varepsilon$.

### C.2 UNSUPERVISED LATENT-SPACE EDITS OF GANS

Voynov & Babenko (2020) learn a matrix where each column is a direction. Yet, Voynov & Babenko (2020) require a human to interpret the learnt directions. To avoid bias during the selection of directions in our comparison, we use the pre-trained attribute regressor to automatically identify the most significant directions. Concretely, we first generate 100 images and edit them with target attributes of various degrees. Next, we use the regressor $R$ on the edited images to predict attributes. We choose for an attribute edit the direction on which all the edited images have the overall highest changing response on the target attribute.

## D.    ADDITIONAL ANALYSIS

### D.1 GAN INVERSION PERFORMANCE

We use an optimization-based GAN inversion method (Abdal et al., 2019) to find optimal latent vectors that can best reconstruct the real images via the generator. To examine the effect of GAN inversion performance on our approach, we terminated the GAN inversion approach at different training steps, i.e., 500 and 4,000 iterations. We show averaged reconstruction MSE loss in Tab. 3. In this case, we reconstructed 20 real face images and edited their "Smile" and the "Blond hair" attribute with 10 different degrees $\varepsilon$, i.e., 200 images in total for each attribute editing. Quantitative evaluation on image identity preservation and numerical changes on the other semantically independent attributes for the reconstructed face images are given in Tab. 4. The results in Tab. 4 suggest that the performance of the GAN inversion method affects our method to some degree. Visualized inversion and editing results are shown in Fig. 10. The qualitative results suggest that our method still works remarkably well on the worse inversion image.

Table 3: **Averaged MSE loss of reconstructing 20 real face images.** The GAN inversion method (Abdal et al., 2019) was trained and terminated at 4k and 500 iterations with averaged MSE loss in the right column.

| Training iterations | MSE |
|---|---|
| 4k iters | 1657.40 |
| 500 iters | 3096.75 |

### D.2 ABLATION STUDY

We conduct an ablation study with single- and joint-distribution training strategies in our method. Specifically, single-distribution sampling refers to learn one attribute direction at a time. In contrast, joint-distribution training means to train multiple attribute directions simultaneously. Fig. 11 shows the visualized results with the two training strategies on the face and the scene dataset. We observe that the model with the single-distribution training strategy generates more unexpected changes as the manipulation degree is getting large, e.g., darker scene colors by the model trained for a single attribute, shown in Fig. 11 (b).

Table 4: **Quantitative evaluation of identity preservation (ID) and attribute changes (Attr)**. We edited two attributes for the real face images, i.e., "Smile" (col.2-7) and "Blond hair" (col.8-13).

| | Smile | | | | | | Blond hair | | | | | |
| | ID ($\uparrow$) | | | Attr ($\downarrow$) | | | ID ($\uparrow$) | | | Attr ($\downarrow$) | | |
| $|\hat{\varepsilon}|$ | (0, .3] | (.3, .6] | (.6, .9] | (0, .3] | (.3, .6] | (.6, .9] | (0, .3] | (.3, .6] | (.6, .9] | (0, .3] | (.3, .6] | (.6, .9] |
|---|---|---|---|---|---|---|---|---|---|---|---|---|
| 4k iters | **.997** | **.994** | **.986** | **.077** | **.108** | **.147** | **.988** | **.979** | **.94** | **.1** | **.112** | **.16** |
| 500 iters | .996 | .99 | .982 | .087 | .138 | .165 | .978 | .936 | .914 | .122 | .158 | .185 |

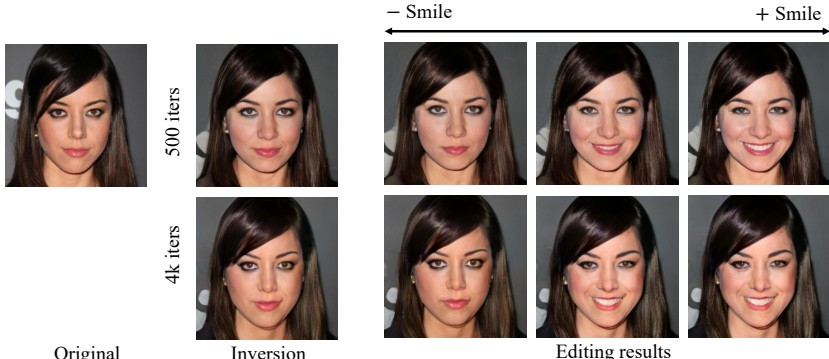

Figure 10: **Visual comparisons of "Smile" editing with different reconstructed images**: (col.1) Original image; (col.2) reconstructed images with 500 (top) and 4k (bottom) inversion optimization steps; (col.3-6) results of editing the "Smile" attribute.

## E.   ADDITIONAL RESULTS

### E.1   SCENE IMAGE EDITS ON STYLEGAN2

We show more results of our method on continuous scene image edits in Fig. 12 - 15. The semantically edited attributes include "clouds" (Fig. 12), "brightness" (Fig. 13), "snow" (Fig. 14), and "summer" (Fig. 15).

## F.   BROADER IMPACT

From an application perspective, our method is effective and efficient with regard to image manipulation and photo-realism, which we hope will contribute to 2D and 3D controllable image editing tasks. Moreover, usage of deep nets to learn mappings between spaces is still not well understood, e.g., mappings between low-dimensional space and image space in classification and generation tasks. We hope our method provides inspiration for representation learning and a first step for a new view with regard to deep net interpretability.

Obviously, we are aware of the dangers of automated image manipulation. Similar to deepfake tasks whose aim is to produce fabricated images and videos that appear to be real, improper use of image manipulation approaches might raise negative issues with regard to information security, property, etc. Beyond that, edited image detection techniques Wang et al. (2019a;b) are proposed recently to avoid the aforementioned issues, which promotes growth in both domains.

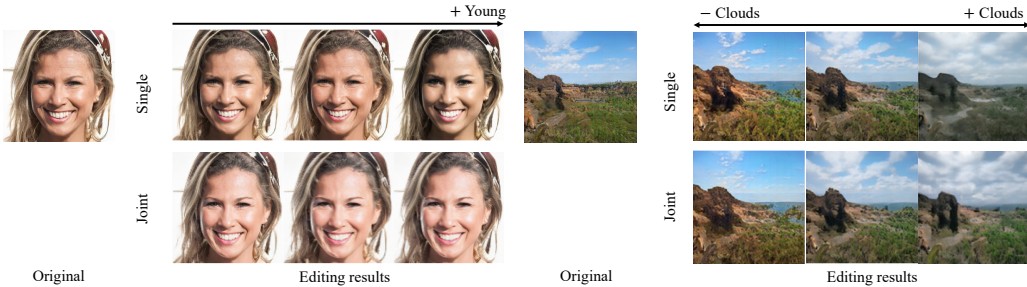

(a) Visual comparisons of single- and joint-distribution sampling on the face dataset

(b) Visual comparisons of single- and joint-distribution sampling on the scene dataset

Figure 11: **Visual comparisons of single- and joint-distribution training.** Single-distribution training refers to train one attribute direction at a time, while joint-distribution training means to train multiple attribute directions simultaneously.

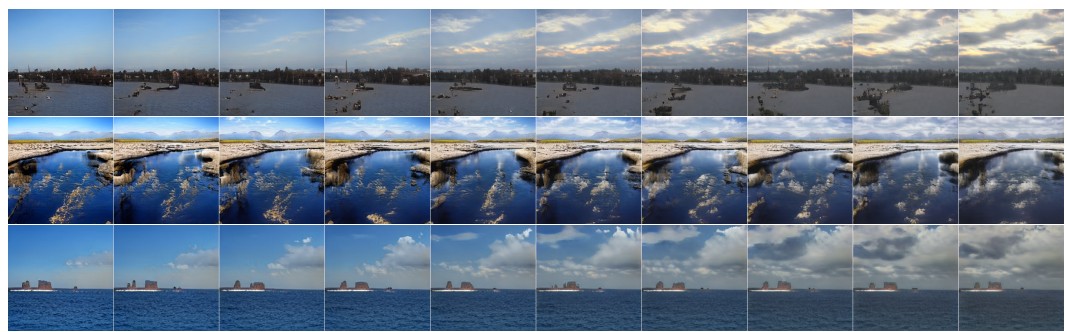

Figure 12: **Additional results.** Continuous image edits using StyleGAN2 (Karras et al., 2019b) on the "clouds" attribute.

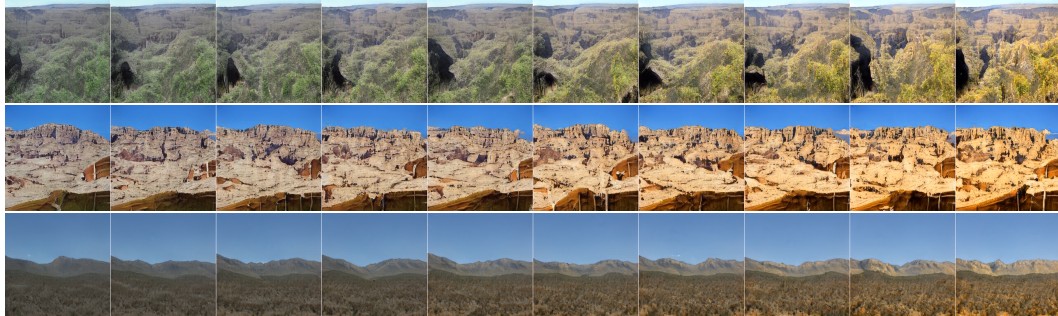

Figure 13: **Additional results.** Continuous image edits using StyleGAN2 (Karras et al., 2019b) on the "brightness" attribute.

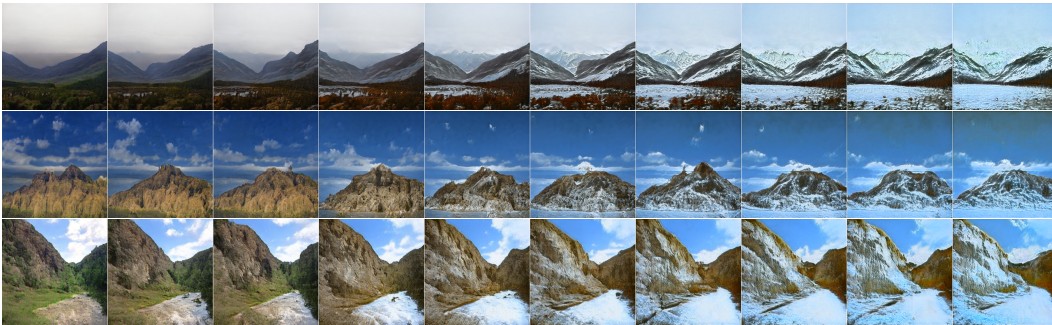

Figure 14: **Additional results.** Continuous image edits using StyleGAN2 (Karras et al., 2019b) on the "snow" attribute.

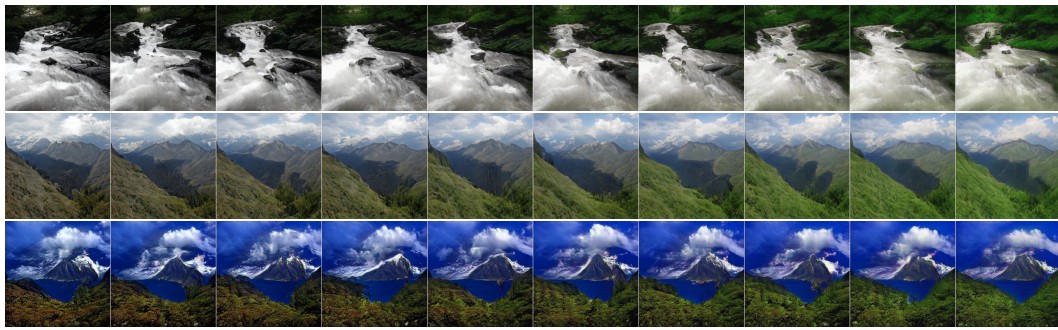

Figure 15: **Additional results.** Continuous image edits using StyleGAN2 (Karras et al., 2019b) on the "summer" attribute.

