# OpenReview forum: "Enjoy Your Editing: Controllable GANs for Image Editing via Latent Space Navigation"
_ICLR.cc/2021/Conference — ICLR 2021 Poster_

### Official Review · AnonReviewer1 · 2020-10-26
**claims are not well validated**

**Rating:** 6
**Confidence:** 3

**Review:**

In this paper, the authors propose an image attribute editing method by manipulating the GAN latent vector. Specifically, this paper uses a pre-trained GAN to synthesize images, a pre-trained regressor to get the image attributes, and trains a network T to find meaningful latent-space directions. It then edits image attributes by modifying the input latent vector, described as z' = z + T(z)ε. The experimental results show that the proposed method performs better than other selected methods to some degree.

Strengths:
1) The idea of controllable editing is intuitive and interesting.
2) Quantitative results on face datasets (Tab. 1 and Tab. 2) show its superiority over other selected methods.
3) Visualization results on both the natural scene and face datasets show its effectiveness on the parts to be edited (e.g. clouds for “remove clouds” and mouth for “add smile”).

Weaknesses:
1) Visualization results on natural scene datasets show that its ability to maintain image identity needs to be improved. For example, the image identities in Fig. 3 are changed during the editing, like the mountains' shapes in row 1 and trees' shapes in row 3.
2) The authors claim that their edits are disentangled, but the visualization results on face datasets don’t support this point very well. For example, the baselines unexpectedly add "glasses" when aging the face in Fig. 4. However, the proposed method also adds glasses on the man's right eye (figure on the bottom right corner).
3) Lack of comparisons of continuous image editing with other methods.

---

> ### Author Response · Authors · 2020-11-20
> **Response to Reviewer 1**
>
> We thank Reviewer 1 for the valuable feedback.
>
> 1. **" Visualization results on natural scene datasets show that its ability to maintain image identity needs to be improved. For example, the image identities in Fig. 3 are changed during the editing, like the mountains' shapes in row 1 and trees' shapes in row 3."**
>
> We agree that results could be improved in multiple ways, including better image identity preservation. Nonetheless, we’d like to mention that prior related work (Shen et al. 2020, Voynov et al. 2020) didn’t consider image identity preservation at all. In contrast, we provide a first attempt to explicitly maintain the identity which is according to our opinion successful to a reasonable degree. Supporting results are shown in the manuscript, including Fig. 1, 3-6, and Tab. 1-2.
>
> 2. **"The authors claim that their edits are disentangled, but the visualization results on face datasets don’t support this point very well. For example, the baselines unexpectedly add "glasses" when aging the face in Fig. 4. However, the proposed method also adds glasses on the man's right eye (figure on the bottom right corner)."**
>
> We don’t think our method adds glasses on the man’s right eye in Figure 4. Instead, wrinkles on the face are increased. Beyond understanding bias from a variety of people, Figure 4 suggests that our approach outperforms the representative baseline approaches, for which results are provided in Figure 4 (a) and (b).
> We also provided more supporting evidence for our disentanglement claim in Figure 1, Figure 6, and Table 1-2. For example, in Figure 6, we showcase better results than the state-of-the-art supervised method (Shen et al., 2020) for editing multiple (2~3) attributes simultaneously. This suggests that the method can achieve better disentanglement and image identity preservation. Similarly, Table 1-2 supports the claim quantitatively.
>
> 3. **"Lack of comparisons of continuous image editing with other methods."**
>
> In the manuscript, we compared our method to a representative continuous image-to-image translation approach, RelGAN (Wu et al., 2019). For example, Figure 3 (c) and Figure 14 (c) present a comparison to this approach on the Scene dataset. Specifically, as shown in Figure 3, we notice that this baseline didn’t perform well with regard to editing image details such as ``removing clouds’’.
>
> **References:**
>
> RelGAN: Multi-Domain Image-to-Image Translation via Relative Attributes, Po-Wei Wu et al., In ICCV 2019.
> Interpreting the Latent Space of GANs for Semantic Face Editing, Yujun Shen et al., In CVPR 2020.
> Unsupervised discovery of interpretable directions in the gan latent space, Andrey Voynov and Artem Babenko, In ICML, 2020.

---

### Official Review · AnonReviewer2 · 2020-10-28
**This work produces very interesting results, but the manuscript seems to need substantial revisions.**

**Rating:** 6
**Confidence:** 4

**Review:**

This paper presents a new approach for the semantic image editing task by allowing the controllable transformation on the latent space. Authors proposed to integrate an attribute regression network for training the transformation functions. The local transformation T is learned from a simple MLP conditioned on the latent vector z. Two outputs of the regression module for the original latent vector z and the transformed one z+T*epsilon are used to minimize the cross-entropy loss. Experiments validate the effectiveness of the proposed method in terms of manipulation quality.

* Pros
1) Local transformation function T would be a more appropriate choice for transforming the latent vector.

* Cons
1) The manuscript contains lots of vague parts.
- Generator G, discriminator D, and regressor R are all pre-trained. How did you obtain the pre-trained regressor R?
- ‘Joint-distribution sampling and training’ part in Section 3.2 is hard to understand. More details would be needed, including its practical implementation and pros over the separate binary sampling.
- This method requires using inversion results. Does the accuracy of the inversion results affect the performance of the proposed method?

2) An ablation study on the joint-distribution sampling would be needed to validate their effectiveness.

3) It seems that editing control (e.g., +Night or +Snow) is related to how to sample the epsilon vector as shown in Figure 2. Did you use the sampling strategy that is adaptive with respect to the given editing control? For instance, when editing images into a night scene, how do the method sample the epsilon vector? It seems that there is no such functionality in Algorithm 1. Please revise the manuscript by specifying this part.

4) Table 2 seems to measure the cosine similarity, while Table 1 evaluates the change of the attributes. How did you calculate the change of the attributes?

5) The results in Figure 8 (a) are based on the global transformation T with the proposed losses. Did you obtain these results using MLP direction functions or simple linear-layer direction functions?

---

> ### Author Response · Authors · 2020-11-20
> **Response to Reviewer 2**
>
> We thank Reviewer 2 for the valuable feedback.
>
> 1. **" How did you obtain the pre-trained regressor R?"**
>
> We mentioned details of the pre-trained regressor R in Section 4, Implementation details, line 6-10: the regressor R is a modified ResNet-50 whose last linear layer is replaced with an output dimension of 40 (the number of attributes in our dataset). We trained R for 500 epochs and used the weights with the best val/test MSE on the datasets.
>
> To be more specific, for example, on the scene dataset (Laffont et al. 2014), R has the minimum validation set MSE of 0.0139 at epoch 384. The MSE scores provided in the original paper (Laffont et al. 2014) are 0.018 (by SVR), 0.045 (by SVM), and 0.063 (by logistic regression).
>
>  2. **"Details of ‘Joint-distribution sampling and training’ part in Section 3.2 would be needed."**
>
> Thanks for suggesting, we’ll add a description in the manuscript. We also summarize the sampling and training strategies in the following. Take a global transformation matrix $T$ as an example, the transformation equation for the latent vector is  $z’ = z + T \epsilon$.
>
> * $\epsilon$ is an N-dimensional vector.
> * $T = [d_1, …, d_N]$, where $d_i$ is a vector that has the same dimension as the latent vector $z$, and $i \in \{1,..., N\}$.
> * The regressor R estimates all attribute values for image $G(z’)$ during training. Let $\hat{\alpha}’ = R(G(z’))$, where $\hat{\alpha}’$ is also an N-dimensional vector.
>
> Given $d_i$ and the N-dimensional $\epsilon$ and $\hat{\alpha}’$, we are able to modify multiple directions at once using the training algorithm shown in APPENDIX-A. In contrast, prior state-of-the-art work (Shen et al. 2020) employs a scalar $\epsilon$, and finds the directions one at a time, i.e., N times in total.
>
> 3. **"Does the accuracy of the inversion results affect the performance of the proposed method?"**
>
> We added experiments in the supplementary material to study how GAN inversion impacts results. In the supplementary material, Tab.4 and Fig.18 present quantitative and qualitative results on real face images, comparing different GAN inversion qualities. Tab.4 indicates that GAN inversion quality effects results to some degree, which isn’t entirely surprising. Nonetheless, Fig.18 suggests that we can still edit the attributes well on inverted images with larger reconstruction loss.
>
>
> 4. **"An ablation study on the joint-distribution sampling would be needed to validate their effectiveness."**
>
> We summarize the effectiveness of our joint-distribution sampling strategy into two points:
>
> * Our joint-distribution sampling and training strategy is primarily compared to the prior state-of-the-art work (Shen et al. 2020) that has to find one direction at a time without considering the other attributes. Therefore, one advantage of such a joint-distribution sampling and training is that we can obtain multiple directions within one training iteration. Moreover, our results in the paper suggest better results of our method compared to the prior state-of-the-art work (Shen et al. 2020) with respect to identity preservation and attribute manipulation.
>
> * In the supplementary material, we also added a comparison between joint-distribution sampling and a single-distribution training strategy of our approach on both face and scene data. Results in Fig.19 show that the model with the single-distribution training strategy may generate more unexpected changes as the manipulation degree is getting large, e.g., darker scene colors by the model trained on a single attribute.
>
>
>
> 5. **"Specify the sampling strategy of the epsilon vector as shown in Figure 2."**
>
> We mentioned the sampling strategy of the epsilon on page 4, in the paragraph above the Object function paragraph. An epsilon is drawn from a distribution uniform in $[-1, 1]^N$ while considering the constraint that the sum of the attribute values and the epsilon is in the range [0,1].
>
> 6. **"Table 2 seems to measure the cosine similarity, while Table 1 evaluates the change of the attributes. How did you calculate the change of the attributes?"**
>
> We used a straightforward strategy to calculate the changes of the attributes: editing images on one attribute ideally shouldn’t change the other irrelevant attributes. That is to say, the other attribute values of an image before and after editing,  predicted by R, should be unchanged. Hence, we calculated and averaged the absolute difference on each of the other attribute values, predicted by R, for an image before and after editing, to measure the attribute changes.
>
>
> 7. **"The results in Figure 8 (a) are based on the global transformation T with the proposed losses. Did you obtain these results using MLP direction functions or simple linear-layer direction functions?"**
>
> The global transformation in Figure 8 is a linear-layer direction independent of the inputs. We mention that such a global transformation setting is widely used in the prior work (Shen et al. 2020; Voynov et al. 2020).

---

### Official Review · AnonReviewer4 · 2020-11-01
**This paper presented a latent-space editing framework for semantic image manipulation. The pre-training of regressor R should be given in more details.**

**Rating:** 6
**Confidence:** 4

**Review:**

This paper presented a latent-space editing framework for semantic image manipulation. The idea is interesting and plausible, and experiments also show its effectiveness. However, I still have some concerns:

1. The pre-training of regressor R should be given in more details.
2. The results of the proposed method heavily depends on the GAN inversion. However, it can be seen from Fig. 6 that some details are still lost by GAN inversion.
3. In term of simultaneous multiple attribute transformations, it is interesting to refer to the following reference to guarantee the latent consistency in the proposed framework:
[r1] Inducing Optimal Attribute Representations for Conditional GANs, ECCV 2020.

---

> ### Author Response · Authors · 2020-11-20
> **Response to Reviewer 4**
>
> We thank Reviewer 4 for the valuable feedback.
>
> 1. **"The pre-training of regressor R should be given in more detail."**
>
> We mentioned details of the pre-trained regressor R in Section 4 Implementation details, line 6-10: the regressor R is a modified ResNet-50 whose last linear layer is replaced with an output dimension of 40 (the number of attributes in our dataset). We trained R offline for 500 epochs and used the weights with the best val/test MSE on the datasets.
> 	To be more specific, for example, on the scene dataset (Laffont et al. 2014), our R has the minimum validation MSE of 0.0139 at epoch 384. The MSE scores provided in the original paper (Laffont et al. 2014) are 0.018 (by SVR), 0.045 (by SVM), and 0.063 (by logistic regression).
>
> 2. **"The results of the proposed method heavily depend on the GAN inversion. However, it can be seen from Fig. 6 that some details are still lost by GAN inversion."**
>
> We agree, there is a trade-off between editing within image space and editing in the latent space of a GAN. An advantage of image editing in the latent space of a GAN: discovered latent variable manipulations permit continuous semantic image edits for multiple pixels using a lower-dimensional space. However, current latent-space editing approaches surely depend on GAN inversion for real image editing. In this work, we mainly focus on finding a more controllable navigation strategy for image editing. Nonetheless, we notice that multiple GAN inversion methods (Fang et al. 2019; Zhu et al. 2020) were proposed recently, which may help to further improve the results.
>
> 3. **"In terms of simultaneous multiple attribute transformations, it is interesting to refer to the following reference to guarantee the latent consistency in the proposed framework: [r1] Inducing Optimal Attribute Representations for Conditional GANs, ECCV 2020."**
>
> Thanks a lot, we appreciate the reference to more literature. We already cited it in the revised version.
>
> **References:**
>
> Transient attributes for high-level understanding and editing of outdoor scenes, Pierre-Yves Laffont et al., In TOG, 2014.
> Co-Generation with GANs using AIS based HMC, Tiantian Fang et al., In NeurIPS 2019.
> In-Domain GAN Inversion for Real Image Editing, Jiapeng Zhu et al., In ECCV 2020

---

### Official Review · AnonReviewer3 · 2020-11-09
**Recommend to a clear accept**

**Rating:** 8
**Confidence:** 4

**Review:**

#### Summary:
The paper proposes a new simple, yet powerful and alternative method of editing the semantic attributes of images generated using pre-trained GAN models as well as a pre-trained regressors. The approach allows for the manipulation of single or multiple various image attributes, while preserving the identity of the original image in contrast to the baseline method of Shen et. al 2019. The method focuses on the manipulation of the latent space, in contrast to the popular image space editing methods.

The paper is easy to read and understand. The authors have presented their method and analysis which are clear and understandable, supported quite well with the provided examples and figures.

#### Strengths:
Experiments are conducted on various datasets and compared to various methods - supervised and unsupervised
Their method is able to consistently preserve the image content (in case of scenes) and identity (in case of faces)
Showing inversion results along the transformation shows the robustness and tractable nature of their method.

#### Weakness:
It is not entirely clear how the local transformation are discovered for different z values independently, or rather what the difference is in discovering the global transformation vs discovering the local transformation


#### Questions to the Authors:
1. In comparison with the Voynov & Babenko model, how many of the directions that their model was able to find did you look at?
2. What is the accuracy / results of the performance metric of the different used pre-trained regressors? Given that the perceptual losses were used while training, the regressors’ performance values could shed some light in this direction.

---

> ### Author Response · Authors · 2020-11-20
> **Response to Reviewer 3**
>
> We thank Reviewer 3 for the valuable feedback.
>
> 1. **"It is not entirely clear how the local transformations are discovered for different z values independently, or rather what the difference is in discovering the global transformation vs discovering the local transformation."**
>
> A global transformation refers to the direction primarily used in (Hahanian et al. 2019; Shen et al. 2020; Voynov et al. 2020), which is a vector representing the path in the GAN latent space, whereas a local transformation is a function with its output depending on the input latent vectors. We will clarify this in the main manuscript.
>
> 2. **"In comparison with the Voynov & Babenko model, how many of the directions that their model was able to find did you look at?"**
>
> The official repository of  Voynov's method (https://github.com/anvoynov/GANLatentDiscovery) provided their annotated meaningful directions.
> * 13 annotated directions of StyleGAN2 on the face dataset, e.g., redness, saturation, luminance, gender, etc.
> * 6 annotated directions of Progressive GAN on the face dataset, e.g., zooming and rotating.
>
> We also visualized 20 examples for each of their latent directions of StyleGAN2 on the face dataset. Each example was edited with 10 different step sizes uniformly sampled from [-6, 6] (suggested in the paper). The number of semantically meaningful directions we looked at is identical to what they provided.
>
> 3. **"What is the accuracy / results of the performance metric of the different used pre-trained regressors? Given that the perceptual losses were used while training, the regressors’ performance values could shed some light in this direction."**
>
> Currently, our regressors have a relatively small MSE, e.g., 0.0139 of validation MSE on the scene dataset. We didn’t explore the effects of different regressors.
>
> **Reference:**
>
> On the "steerability" of generative adversarial networks, Ali Hahanian et al., In ICLR 2019.
> Interpreting the Latent Space of GANs for Semantic Face Editing, Yujun Shen et al., In CVPR 2020.
> Unsupervised discovery of interpretable directions in the gan latent space, Andrey Voynov and Artem Babenko, In ICML, 2020.

---

### Decision · Program_Chairs · 2021-01-07
**Final Decision**

**Decision:**

Accept (Poster)

**Comment:**

All the reviewers rate the paper above the bar. They like the experiment results and think the proposed latent space editing approach makes intuitive sense. While several weakness points were raised, including a lack of continuous editing comparison and sometimes vague descriptions, they were not considered major to reject the paper. After consolidating the reviews and rebuttal, the AC agrees with the reviewer assessment and recommends accepting the paper.